# RETRACTED: Is There an Overlap Between Eating Disorders and Neurodevelopmental Disorders in Children with Obesity?

**DOI:** 10.3390/nu11102496

**Published:** 2019-10-17

**Authors:** Elisabet Wentz, Anna Björk, Jovanna Dahlgren

**Affiliations:** 1Department of Psychiatry and Neurochemistry, Institute of Neuroscience and Physiology, University of Gothenburg, 40530 Gothenburg, Sweden; 2Department of Pediatrics, Institute of Clinical Sciences, University of Gothenburg, 40530 Gothenburg, Sweden; anna.bjork.2@gu.se (A.B.); jovanna.dahlgren@gu.se (J.D.)

**Keywords:** eating disorders, loss of control eating, obesity, attention-deficit/hyperactivity disorder, autism spectrum disorder, body mass index (BMI)

## Abstract

This study aimed at assessing the prevalence of eating disorders (EDs) and ED symptomatology in children with obesity, and at investigating whether EDs occur more often among individuals with a comorbid attention-deficit/hyperactivity disorder (ADHD) and autism spectrum disorder (ASD). Seventy-six children (37 girls, 39 boys, age 5–16 years) were recruited at an outpatient obesity clinic. The adolescents completed ED instruments including The Eating Disorder Examination Questionnaire (EDE-Q) and The Eating Disorder Inventory for children (EDI-C). The parents of all participants were interviewed regarding the child’s psychiatric morbidity. Diagnoses of ADHD and ASD were collected from medical records. Anthropometric data were compiled. Eight participants (11%) fulfilled the criteria for a probable ED and 16 participants (21%) had ADHD and/or ASD. Two adolescent girls had a probable ED and coexistent ADHD and ASD. No other overlaps between EDs and ADHD/ASD were observed. Loss of control (LOC) eating was present in 26 out of 40 (65%) adolescents, seven of whom had ADHD, ASD or both. LOC eating was not overrepresented among teenagers with ADHD and/or ASD. Weight and shape concerns were on a par with age-matched adolescents with EDs. EDs and ED behavior are more common among children/adolescents with obesity than in the general population. There is no substantial overlap between EDs and ADHD/ASD in adolescents with obesity.

What is already known about this subject:Loss of control (LOC) eating is overrepresented in overweight/obese children;Attention-deficit/hyperactivity disorder (ADHD) and autism spectrum disorder are more prevalent in children and adolescents with obesity than in peers without obesity;Both LOC eating and ADHD are conditions characterized by impulsivity.

What this study adds:Eating disorders with a binge eating symptomatology were overrepresented among children and adolescents with obesity;The majority of adolescents with obesity had weight and shape concerns in parity with adolescents with eating disorders;Two thirds of the adolescents with obesity reported concurrent LOC eating, regardless of whether or not they met the criteria for ADHD or autism.

## 1. Introduction

Obesity in children, adolescents and adults is perhaps the most salient global health problem of our time. The number of children and adolescents worldwide with obesity was estimated to be 110 million in 2013, and the age-standardized prevalence was 5% among children, with much higher numbers in some countries (range 2%–30%) [1].

According to the fifth edition of the American Psychiatric Association’s Diagnostic and Statistical Manual of Mental disorders (DSM-5) [2], the onset of traditional eating disorders (EDs), including anorexia nervosa, bulimia nervosa and binge-eating disorder (BED), normally occurs during adolescence or in adulthood. An association between BED and obesity was first reported among adult individuals in the 1990s, when a field trial found BED to be more common in individuals in weight control programs [3]. BED is rare in children and adolescents in the general population. However, one of the BED symptoms, namely loss of control (LOC) eating, can be recognized in children. LOC eating is an experience of loss of control over how much or what one eats, regardless of the amount of food consumed [4]. In a recent meta-analysis, one quarter of children and adolescents with overweight and obesity engaged in binge or LOC eating [5].

Apart from EDs, other psychiatric disorders and symptoms can contribute to the development of overweight and obesity in childhood. Attention-deficit/hyperactivity disorder (ADHD), a psychiatric disorder characterized by symptoms of inattention, impulsivity and hyperactivity with childhood onset, has been found to be a risk factor for obesity in both children [6,7] and adults [8,9]. LOC eating is probably the most common ED in children with ADHD. One study reported a twelvefold increased rate in LOC eating among children with ADHD compared with children without ADHD [10]. Children with overweight/obesity and LOC eating were more likely to have comorbid ADHD than children with overweight/obesity without LOC eating. The study focused on the overlap between ADHD and LOC eating, while other neurodevelopmental disorders (including autism) and other ED diagnoses (e.g., BED) were not considered [10].

ADHD is characterized by impulsive behavior. BED and LOC eating also exhibit a lack of inhibition [11]. According to a review by Schag and co-workers [12], there are two aspects to impulsivity that are more salient in individuals with obesity and coexistent BED/LOC eating, as opposed to individuals with obesity alone. One of the impulsive behaviors is called “reward sensitivity” (drive to obtain rewarding stimuli) and the other is labeled “rash-spontaneous behavior” (“hasty behavior without recognizing the consequences”). People with obesity but without BED are able to inhibit food intake more readily than people with obesity and co-existent BED. Schag’s group also concluded that BED may indicate a more neurobehavioral variant of obesity [12]. It is therefore important to examine whether there is an overlap between different neurodevelopmental disorders, including both ADHD and autism, and EDs in children with obesity.

The population of children and adolescents with autism spectrum disorder (ASD) shows an overrepresentation of extremely high body mass index (BMI) [13]. Weight gain is a common side effect of antipsychotics, which are widely prescribed to children and adolescents with ASD and coexistent behavioral problems [14]. We have previously found an increased rate of ASD, corresponding to 13%, in a clinical setting for children with obesity [7].

At the Obesity Center at the Queen Silvia Children’s Hospital in Gothenburg, Sweden, we examined 76 children and adolescents with obesity regarding EDs and associated cognitive and behavioral symptoms. A previous publication based on the same sample indicated that a variety of neurodevelopmental disorders—ADHD and ASD, in particular—are prevalent in children with obesity [7]. The hypothesis in the present study was that EDs and associated symptoms are overrepresented among children and adolescents with obesity. We also hypothesized that EDs that include impulsive eating behavior (i.e., binge eating/LOC eating and purging behavior) were more common in children and adolescents with obesity and coexistent ADHD and/or ASD, compared with children and adolescents with obesity without ADHD and ASD.

## 2. Methods

### 2.1. Participants

Seventy-six children and adolescents (37 girls and 39 boys) were recruited at the Obesity Center at the Queen Silvia Children’s Hospital in Gothenburg, Sweden. In accordance with the inclusion criteria, the participating children had all been recently referred to the unit, were 5–18 years old and both the parents and the patient were fluent in Swedish. In total, 134 children and adolescents were invited to participate in the study and 58 declined participation. Those who declined differed from those who took part only in terms of fasting glucose levels, where those who declined had statistically significantly higher levels [7].

### 2.2. Instruments

Two ED instruments, the Eating Disorder Examination Questionnaire (EDE-Q) [15] and the Eating Disorder Inventory for children (EDI-C) [16], were used in the study since they assess different aspects of EDs. The EDE-Q measures the core symptoms of EDs and can be used as a diagnostic tool, while the EDI-C assesses typical cognitive and behavioral characteristics observed in individuals with EDs.

The Eating Disorder Examination Questionnaire (EDE-Q) is a self-report questionnaire, which consists of 36 items and generates the four subscales, “Dietary restraint”, “Eating concern”, “Shape concern” and “Weight concern” [15]. Items measuring binge eating, self-induced vomiting and LOC eating are also incorporated in the scale, but not included in the total score. Swedish cutoff scores for EDs [17], matched for age and gender, were used for comparison. LOC eating was measured by using the question: “How many of these times did you have a sense of loss of control over your eating (at the time when you were eating)?”. Referring to the previous question: “Over the past 28 days, how many times have you eaten what other people would regard as an unusually large amount of food (given the circumstances)?”. In accordance with a classification suggested by Schlüter et al. [18], recurrent LOC eating was classified as >4 episodes in the last month.

The Eating Disorder Inventory for children (EDI-C) [16], a self-report questionnaire dealing with symptoms of ED and related behavior, was only administered to the children and adolescents between 10 and 17 years of age, among the total study population of 76 individuals. The instrument consists of 91 items subdivided into eleven subscales. The first three subscales, “Drive for thinness” (excessive concern with dieting), “Bulimia” (uncontrolled overeating), and “Body dissatisfaction” (discontent with body shape), are categorized as symptom subscales. The remaining eight are classified as psychological subscales and include “Ineffectiveness” (worthlessness and lack of control over one’s life), “Perfectionism”, “Interpersonal distrust” (reluctance to have close relationships), “Interoceptive awareness” (uncertainty in respect of emotional states related to hunger and satiety), “Maturity fears”, “Asceticism” (seeking virtue in self-discipline), “Impulsive regulation”, and “Social relationships” (insecurity in terms of social interplay). A total score can be calculated based on the sums of the eleven subscales. Swedish norms have been published [19] and include data for the age groups. The psychometric properties of the EDI-C and EDI-2 have been shown to be comparable for adolescent girls with EDs (Cronbach’s alphas 0.70–0.91) [19]. For Swedish girls aged 13–17 years with an ED, the norms also include suggested cutoffs for the total score and for all the subscales [19].

The Development and Well-Being Assessment (DAWBA) [20] was used in 73 of the 76 children. The DAWBA is a parental interview form pertaining to psychiatric diagnoses during pediatric years, according to the fourth edition of the American Psychiatric Association’s Diagnostic and Statistical Manual of Mental Disorders DSM-IV [21]. The DAWBA domains used in the present study covered anorexia nervosa, bulimia nervosa, ADHD, tic disorders and major depression (MD). The BED diagnosis is not included in the DAWBA, since BED is a diagnosis that was introduced in 2013 when the DSM-5 was published [2]. The DAWBA interview lacks items assessing the associated BED symptoms including feelings of disgust or guilt, eating alone because of feeling embarrassed, and eating rapidly, which prevented us from assigning definite BED diagnoses. “Probable BED” was assigned when the parents reported episodes of binge eating, including LOC eating, at least once a week for three months, without any inappropriate compensatory behavior, including self-induced vomiting and misuse of laxatives and diuretics. At the time of the study, the DSM-5 had not yet been published and the DAWBA was therefore considered the best instrument to assess EDs.

The neurodevelopmental disorders of ADHD and ASD were assigned based either on medical records or the DAWBA (only pertaining to ADHD) (for details, see also Wentz et al. [7]).

### 2.3. Ethical Approval

The study was approved by the regional ethical review board in Gothenburg, Sweden (718-09). Written informed consent was obtained from both children and parents.

### 2.4. Statistical Analyses

BMI is presented as the standardized deviation score (SDS) in order to compare patients of different ages. Means, standard deviations (SDs) and ranges were calculated for descriptive purposes. Non-parametric tests were used, due to scale scores not being normally distributed. To analyze differences between groups (individuals with or without an ED, individuals with or without LOC eating) in terms of continuous variables (BMI SDS and age), the Wilcoxon (Mann–Whitney) rank sum test was performed. The Chi-squared χ^2^ tests (including Fisher’s exact test and continuity correction) were used to compare categorical data, including (1) EDs in individuals with and without ADHD/ASD; (2) LOC eating in individuals with and without ADHD/ASD; (3) depression in individuals with and without EDs, and (4) LOC eating in males and females. Statistical significance was defined as *p* < 0.05. All data were analyzed using IBM SPSS version 24 (IBM Corp, Armonk, NY, USA).

## 3. Results

Anthropometric data including gender, age and BMI SDS are presented in Table 1.

### 3.1. Psychiatric Disorders Including EDs, ADHD, ASD and Major Depression

Eight participants were classified as having a probable ED. Four had probable BED (3 girls, 1 boy), and four (2 girls, 2 boys) had a milder variant of the same disorder, “Other specified feeding or eating disorder, Binge eating disorder”, (OSFED-BED), according to the DSM-5 (see Table 2). Sixteen (21.1%) individuals in the total study group had ADHD, ASD or both, and ten (13.2%) had major depression (Table 2).

### 3.2. Correlations Between EDs and Other Variables

The BMI SDS did not differ between those with an ED and those without an ED (with ED: 3.56, SD: 0.39; no ED: 3.45, SD: 0.59; *p* = 0.164). Coexistent ADHD + ASD were present in two adolescent girls with probable BED and OSFED-BED, respectively. No other person with an ED had comorbid ADHD or ASD. Major depression was not overrepresented among those with an ED (3 out of 8 with ED had major depression; *p* = 0.066).

### 3.3. Prevalence of LOC Eating and Correlations between LOC Eating and Other Variables

Forty individuals completed the EDE-Q and 26 (65%) of them reported LOC eating. There was no difference regarding BMI SDS and age between those who reported LOC eating and those who did not (*p* = 0.130) (Table 1). Nine individuals (22.5%) of those who completed the EDE-Q had ADHD and overlap with LOC eating was present in five of these cases (see Table 2). Nine individuals (22.5%) had ASD and seven of them had comorbid LOC eating. Seven individuals had both ADHD and ASD and all but two reported LOC eating. Altogether, eleven adolescents who completed the EDE-Q had ADHD, ASD or both. Seven of those had LOC eating, corresponding to 63.6%. Of the 26 individuals with LOC eating, 19 (65.5%) had no comorbid ADHD or ASD. LOC eating was not overrepresented among those with ADHD, ASD or both (*p* = 1.00).

### 3.4. EDE-Q

Table 3 shows data from the EDE-Q. A total of 41 out of 47 individuals aged 12–17 years in the study group completed the questionnaire. Altogether, 26 out of 41 (63%) scored above the cutoff for EDs regarding the global score. The subscales “Shape concern” and “Weight concern” were the most salient problems among adolescents, both engaging 66% of the group, while only 32% reported pathological “Eating concern”. Regarding additional data retrieved from the EDE-Q but not included in the subscales, bingeing was most common among girls aged 12–15, affecting seven out of 12 individuals at least once a week, on average (data not shown in Table 2). No adolescents engaged in self-induced vomiting on a regular basis (<one episode of vomiting per 28 days). LOC eating occurred, without any significant gender differences, in 26 of the 40 (65%) adolescents who ticked that item (girls: 13/17; 76%; boys: 13/23; 57%; *p* = 0.32).

### 3.5. EDI-C

In total, 57 out of 60 in the 10–17 year age group in the study group (*N* = 76) completed the EDI-C questionnaire. The EDI-C showed that all age groups with obesity, irrespective of gender, had mean scores above the normative data regarding the three symptom subscales (drive for thinness, bulimia, and body dissatisfaction) (Table 4). Furthermore, girls aged 10–12 scored above the normative data on all the subscales except for asceticism and social insecurity. Boys in the same age group scored below the normative group on all but two psychological subscales (interoceptive awareness and maturity fears).

### 3.6. EDI-C Girls 13–17 Years

Using the Swedish norms for girls with EDs aged 13–17, the scores for the age-matched girls with obesity were above the cutoff for two of the three symptom subscales and six of the eight psychological subscales and above the cutoff for the total scale (Table 5). Body dissatisfaction was the subscale on which most girls with obesity (11/16) scored at or above the cutoff for EDs. A minority (5/16) scored at or above the cutoff for the Bulimia subscale.

## 4. Discussion

This study aimed at investigating the prevalence of EDs, including LOC eating, in a clinical sample of children and adolescents with obesity. The prevalence of EDs (i.e., BED and OSFED-BED) was 10.5%, which is elevated compared with the general adolescent population [23]. Furthermore, the prevalence of LOC eating among adolescents was surprisingly high, engaging two thirds of the teenagers. We further wanted to investigate whether there was an overlap between individuals with ADHD/ASD and individuals with LOC eating. However, the study showed that LOC eating coexisted with obesity irrespective of a concurrent ADHD and/or ASD diagnosis.

BED and the milder variant of the disorder, OSFED-BED, occur in approximately 1%–4% of adults in the general population [24,25]. Prevalence studies of BED and OSFED-BED in children and adolescents are scarce [5]. However, BED and OSFED-BED showed a high prevalence in the present study, which only included children and adolescents with obesity. A previous paper by our group, focusing on neurodevelopmental disorders in the same study group, found both ADHD and ASD to be highly overrepresented [7]. Both individuals with ADHD and ASD have an increased risk of developing comorbid EDs, including anorexia nervosa, bulimia nervosa and BED [26,27,28]. Nevertheless, even if our sample had an increased prevalence of EDs, this was not explained by coexistent ADHD/ASD. LOC eating, a so-called transdiagnostic symptom [29] characteristic of all EDs with binge eating/purging behavior, was extremely common, affecting two thirds of all adolescents in this study. Even so, LOC eating was not primarily neurodevelopmentally related. The ED symptom was frequent, irrespective of whether the individual also had ADHD/ASD or not.

LOC eating was surprisingly common in our teenage sample, engaging two thirds of the group, and the symptom was equally distributed across genders. In a recent review, the prevalence of LOC eating in young people who are overweight and obese is 31.2% [5]. In accordance with our data, the review found no gender differences with regard to LOC eating prevalence. Taken together, both the review and our own data indicate that LOC eating is far more prevalent in young people with obesity, as opposed to community samples reporting a prevalence of approximately 10% [17].

In accordance with Egbert and colleagues [30], we have investigated objective binge eating (corresponding to the diagnosis “probable BED”, based on a structured parental interview). Egbert’s group also assessed a subjective binge eating parameter, defined as LOC eating, where the individual perceived that the amount of food eaten was excessive, but according to clinical ratings the amount of food was not extreme [30]. Their paper showed an association between ADHD and objective binge eating but not with the subjective condition (i.e., an intake of normal amounts of food during LOC eating). The present study could not, however, find an association between ED (objective binge eating) and ADHD. Our study did not include a subjective binge eating parameter. However, a possible association between LOC eating and ADHD was analyzed in our sample, but no relationship was found between the two variables.

Previous studies have shown shared risk factors between obesity and EDs, including body dissatisfaction and weight-related teasing [31]. Both risk factors have been shown to predict overweight and binge eating [32]. Our preadolescent and adolescent participants had a mean score well over the mean scores of the general population in terms of body dissatisfaction. Weight-related teasing was unfortunately not measured in the present study. Our research group had hypothesized that comorbid ADHD and ASD could be a shared risk factor between obesity and EDs, including LOC eating. However, EDs were not overrepresented among those with a neurodevelopmental disorder. On the contrary, we found that the LOC eating symptom was ubiquitous, and not primarily occurring among those with ADHD and/or ASD.

Ro et al. [33] reported that 30%–40% of adult women with obesity scored above the cutoff for the shape and weight concern on the EDE-Q. In our group of adolescents with obesity, the proportion above the cutoff was much higher, involving two thirds of the group. Ro and colleagues [33] found an association with higher scores and younger women, which may explain why the scores in this study were much higher.

There are some limitations of the study that need to be discussed. Firstly, the patients were all tertiary referrals, and several of them had been considered to be treatment-refractory, and therefore referred to a university hospital clinic. Accordingly, the study group cannot be deemed to be representative of children and adolescents with obesity in general, and the many severe cases may bias the results. However, a strength of this study was that EDs and associated cognitive and behavioral symptoms were assessed using a consecutive approach for all new referrals at a university clinic.

Secondly, the sample was relatively small, and some of the statistical calculations would probably not have resulted in trends but in statistically significant differences if the subgroups had been larger (e.g., the tendency towards overrepresentation of comorbid major depression among those with an ED).

Thirdly, no control group was included. However, we had access to Swedish normative data and Swedish cutoff scores for EDs, which enabled us to interpret the severity of the ED symptomatology in the group of young individuals with obesity. Regarding the EDI-C, we compared with Swedish normative and ED data [19]. Swedish age-matched cutoff data were available for the EDE-Q [16].

Fourthly, the instruments used for assessing ED symptoms were only adjusted for children from age 10 (EDI-C) and 12 years (EDE-Q), respectively, and therefore only approximately half the sample could complete the EDE-Q, while 57 individuals were in the age range for completing the EDI-C. We had to divide the patients into subgroups due to different norms for age and gender, resulting in few individuals in each subgroup. With respect to the EDI-C, there are only cutoff data for EDs regarding adolescent girls (i.e., scores above the cutoff are in accordance with how individuals with an ED would score). For this reason, we could not draw any conclusions as to whether preadolescent children and adolescent boys were at risk of developing an ED, since no Swedish cutoff data were available. However, the ED self-report instruments used are established and standardized ED instruments. The EDI-C, which was used in this study, is preferable to the EDI-2 in adolescents, since the statements are addressed to youths and not to adults.

Fifthly, the LOC eating variable was only based on one question from the EDE-Q. However, the individuals had to declare LOC eating behavior at least once a week, on average, during the last 28 days in order to fulfill the LOC eating criterion. The meta-analysis by He et al. [5] observed a lower LOC eating prevalence, corresponding to 13.6%, among studies based on one single question pertaining to LOC eating. Our prevalence data yielded an opposite view, where two-thirds reported LOC eating in spite of only being based on a single item. It should be emphasized that the EDE-Q, encompassing the LOC eating item, is a well-established and widely used questionnaire in ED research. LOC eating was only assessed in adolescents, and therefore we cannot draw any conclusions about LOC eating in those younger than twelve years, and this subgroup made up almost half the sample. Some children/adolescents were upset about the questions in the EDE-Q pertaining to purging behavior and the misuse of laxatives, in particular.

Sixthly, none of the self-report questionnaires regarding ED symptomatology were tailored to children and adolescents with obesity. EDs, including LOC eating, in children and adolescents with obesity have so far been overlooked both in clinical settings and in research [5]. Future research implications should include the development of an ED instrument for this group of young people. Finally, we did not have enough information to assign definite ED diagnoses (i.e., BED and OSFED-BED according to the DSM-5) [2]. The DAWBA interview, for instance, is short on questions focusing on feelings of embarrassment, disgust, depression and guilt in association with binge eating episodes described by the DSM-5. However, a strength of the study is that our clinical ED data were both assessed by experts and based on self-reports.

A strength of this study is that the ADHD diagnoses were either based on information from medical records or assigned using a well-established parental interview format (the DAWBA). Diagnoses of ASD were collected from medical records. With regard to ASD, this diagnosis has, to our knowledge, not previously been reported or examined in combination with traditional EDs in young people with obesity. ASD and ADHD often coexist, and ASD may therefore occur in parallel with ADHD in the children with obesity that we are investigating.

## 5. Conclusions

To conclude, in our clinical population of children and adolescents with obesity, both EDs and LOC eating were overrepresented compared with the general population. ADHD and ASD also exhibited an increased prevalence, but the overlap between EDs and ADHD/ASD was not impressive.

## Figures and Tables

**Table 1 nutrients-11-02496-t001:** Patient characteristics in individuals with and without loss of control eating, expressed as number or mean +/− standard deviation (SD) and range.

	Total Group (*N* = 76)	Assessed for LOC * (*N* = 40)	With LOC (*N* = 26)	Without LOC (*N* = 14)
Gender (male/female)	39/37	23/17	13/13	10/4
Age (years)	12.4 (3.0)	14.3 (1.7)	14.5 (1.7)	14.0 (1.8)
5.1–17.0	11.1–17.0	11.1–17.0	11.3–16.5
Body mass index SDS	3.4 (0.5)	3.4 (0.5)	3.5 (0.4)	3.2 (0.5)
1.9–5.9	1.9–4.3	2.7–4.3	1.9–3.9

* Loss of control (LOC) eating was assessed using the Eating Disorder Examination Questionnaire (EDE-Q). SDS: standard deviation score.

**Table 2 nutrients-11-02496-t002:** Eating disorders, neurodevelopmental disorders and depression in individuals with and without loss of control eating, expressed as number and %.

	Total Group (*N* = 76)	Assessed for LOC * (*N* = 40)	With LOC (*N* = 26)	Without LOC (*N* = 14)
BED	4 (5.3%)	2	1	1
OSFED-BED	4 (5.3%)	4	4	0
ADHD	14 (18.4%)	9	5 **	4
ASD	10 (13.2%)	9	7 **	2
ADHD and/or ASD	16 (21.1%)	11	7 **	4
Major depression	10 (13.2%)	8	6 ***	2

* Loss of control (LOC) eating was assessed with the Eating Disorder Examination Questionnaire (EDE-Q). ** One patient had a coexistent eating disorder. *** Two patients had a coexistent eating disorder. ADHD: attention-deficit/hyperactivity disorder; ASD: autism spectrum disorder; BED: binge eating disorder; OSFED: other specified feeding or eating disorder.

**Table 3 nutrients-11-02496-t003:** EDE-Q results expressed as mean +/− standard deviation (SD) in male and female adolescents (12–15 and 16–17 years old) with obesity.

Variable	Girls with Obesity	Boys with Obesity	Total Group (*N* = 41)
	12–15 years (*N* = 12)	Number above cutoff	16–17 years (*N* = 6)	Number above cutoff	12–15 years (*N* = 17)	Number above cutoff	16–17 years (*N* = 6)	Number above cutoff	Number above cutoff
Global	2.18 (1.02)	7	2.35 (0.88)	3	1.42 (1.18)	11	2.43 (1.19)	5	26 (63.4%)
Restraint	1.72 (1.03)	4	2.03 (1.16)	3	1.25 (1.26)	9	1.2 (0.75)	3	19 (46.3%)
Eating concern	1.08 (1.32)	2	1.30 (1.20)	2	0.60 (0.84)	5	1.93 (1.78)	4	13 (31.7%)
Shape concern	3.17 (1.54)	7	3.42 (2.02)	4	1.92 (1.65)	10	3.33 (1.61)	6	27 (65.9%)
Weight concern	2.73 (1.40)	8	2.63 (1.08)	3	1.91 (1.57)	11	3.25 (1.56)	5	27 (65.9%)

EDE-Q: Eating Disorder Examination-Questionnaire; SD: standard deviation; *N*: number. Cutoff data reference: Ekeroth and Birgegård [16].

**Table 4 nutrients-11-02496-t004:** Eating Disorder Inventory for children (EDI-C) in girls and boys in the age groups 10–12 years and 13–17 years with obesity, expressed as mean +/− standard deviation (SD), compared with age-matched girls and boys in the general population in Sweden.

Subscales	Girls 10–12 Years	Boys 10–12 Years	Girls 13–17 Years	Boys 13–17 Years
	With obesity N = 11	Norms N = 582	With obesity N = 9	Norms N = 372	With obesity N = 16	Norms N = 2046	With obesity N = 21	Norms N = 1698
Drive for thinness	7.00 (8.3)	1.83 (3.7)	2.56 (2.6)	0.63 (2.0)	7.31 (5.2)	3.46 (4.9)	6.00 (5.4)	0.76 (2.3)
Bulimia	1.36 (2.4)	0.37 (1.1)	0.78 (1.4)	0.54 (1.4)	1.69 (3.2)	0.82 (1.8)	2.57 (3.9)	0.88 (2.1)
Body dissatisfaction	9.91 (9.7)	6.04 (6.3)	7.22 (6.6)	3.03 (3.7)	14.69 (9.1)	8.77 (7.4)	14.10 (8.8)	3.61 (4.4)
Ineffectiveness	3.18 (5.5)	3.05 (3.8)	1.22 (1.9)	1.85 (2.8)	6.25 (6.1)	3.20 (4.0)	6.31 (5.8)	1.74 (3.1)
Perfectionism	2.45 (3.1)	2.16 (2.3)	2.56 (1.7)	3.32 (2.6)	4.56 (4.4)	2.59 (2.8)	3.05 (3.1)	3.33 (3.0)
Interpersonal distrust	4.91 (4.3)	3.67 (3.0)	3.00 (2.8)	4.61 (3.0)	4.75 (4.8)	2.93 (3.2)	6.62 (3.8)	4.17 (3.3)
Interoceptive awareness	2.64 (3.1)	1.69 (2.4)	1.78 (2.6)	1.56 (2.1)	6.38 (7.5)	2.82 (3.6)	4.19 (5.0)	1.71 (2.9)
Maturity fears	9.27 (3.8)	7.27 (4.0)	9.00 (3.5)	8.59 (4.3)	7.31 (5.6)	5.46 (3.6)	5.62 (4.1)	6.18 (3.9)
Asceticism	3.73 (3.0)	4.65 (2.9)	3.22 (6.2)	5.82 (3.2)	5.81 (4.2)	5.51 (3.3)	6.33 (4.8)	6.45 (3.3)
Impulse regulation	2.73 (3.0)	2.32 (3.2)	1.33 (1.6)	1.93 (2.9)	4.66 (5.4)	2.97 (3.8)	4.83 (5.8)	2.68 (4.1)
Social insecurity	2.82 (2.6)	3.49 (2.9)	1.67 (1.8)	3.74 (3.2)	4.44 (4.3)	2.94 (2.8)	5.33 (3.4)	3.18 (3.5)
Total	50.00 (31.4)	36.53 (20.2)	34.33 (20.8)	35.61 (14.2)	67.84 (37.4)	41.4 (23.6)	63.88 (38.9)	34.69 (19.8)

Norms: Swedish norms; *N*: number. (Thurfjell et al., 2003, [19]. Thurfjell et al., 2004, [22])

**Table 5 nutrients-11-02496-t005:** EDI-C in girls aged 13–17 years with obesity expressed as mean +/− standard deviation (SD), compared with age-matched girls in the general population and girls with eating disorders.

	Girls with Obesity	Normative Group	ED Group	Cutoff for ED	Number ≥ Cutoff
Number	16	2046	201	N/A	N/A
Drive for thinness	7.31 (5.17)	3.46 (4.91)	11.75 (6.7)	7	9/16
Bulimia	1.69 (3.24)	0.82 (1.82)	3.34 (4.34)	2	5/16
Body dissatisfaction	14.69 (9.12)	8.77 (7.42)	14.47 (7.97)	12	11/16
Ineffectiveness	6.25 (6.08)	3.20 (3.96)	8.47 (6.13)	5	8/16
Perfectionism	4.56 (4.38)	2.59 (2.79)	4.10 (3.59)	3	9/16
Interpersonal distrust	4.75 (4.82)	2.93 (3.16)	4.57 (4.01)	4	9/16
Interoceptive awareness	6.38 (7.49)	2.82 (3.56)	8.40 (6.10)	5	8/16
Maturity fears	7.31 (5.58)	5.46 (3.55)	5.42 (3.76)	5	9/16
Asceticism	5.81 (4.18)	5.51 (3.27)	9.54 (4.82)	7	6/16
Impulse regulation	4.66 (5.35)	2.97 (3.80)	6.83 (5.30)	5	7/16
Social insecurity	4.44 (4.29)	2.94 (2.82)	5.3 (3.82)	4	7/16
Total	67.84 (37.45)	41.4 (23.6)	82.3 (36.9)	57	10/16

EDI-C: Eating Disorder Inventory for children; ED: eating disorder; N/A: not applicable.

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
