# Peer review of "RETRACTED: Is There an Overlap Between Eating Disorders and Neurodevelopmental Disorders in Children with Obesity?"

_nutrients, 2019, doi:10.3390/nu11102496_

Round 1
Reviewer 1 Report
The aim of the present study was to evaluate the prevalence of eating disorders (ED) and ED symptomatology in children with obesity as well as to investigate the frequency in which ED occurs in individuals with a comorbid attention deficit hyperactivity disorder (ADHD) and autism spectrum disorder (ASD). Study involved a clinical sample of 76 children and adolescents with obesity (boys and girls, 5-16 years old) recruited at an obesity outpatient clinic in Sweden. 47 of these children and adolescents aged 12-17 y.o. completed Eating Disorder Examination Questionnaire (EDE-Q); 57 (out of 60) children and adolescents from 10 to 17 y.o. completed the Eating Disorder Inventory for Children (EDI-C) questionnaire; 73 (out of 76) children were subjected to Development and Well-Being Assessment (DAWBA) completed by their parents. Results from the study demonstrated that EDs or behaviors are most common among children or adolescents with obesity compared to the general population and that no substantial overlap between EDs and ADHD/ASD was found in adolescents with obesity.
In my opinion as a peer reviewer of this manuscript, the study is performed well, methods are adequate, data are clearly presented, and I do not find any issues with the statistics. The authors correctly discussed the weaknesses of this study (small sample size, single-centered study, etc.).
Minor technical comment:
- Please define ADHD as attention deficit hyperactivity disorder as first appear in the text and the abstract.
Reviewer 2 Report
line 55: separate age ranges for each: children (2-?) and adolescents (?-19 years of age)
line 57: per cent should be one word
line 57: "much higher numbers in some countries" is vague. I think it may be better to elaborate more
line 60: What is the DSM-5?
line 63: It states that BED is rare in children and adolescents but the previous sentence cites a connection to those that are obese. Are you stating its rare in the general child/adolescent population? If so, please be more specific.
line 70: The first sentence sounds unfinished.
line 69-74: This sounds like the answer to your research question. If that is not the case, explain what these findings were lacking.
line 76-83: I think you should paraphrase more rather than quoting authors directly. It comes off as disjointed.
Also for this paragraph it seems that you are answering your research question with the literature review. Please explain why your research is unique.
line 88: Change "is probably" to "may be" (opinion)
line 99-100: Please rewrite if my interpretation is wrong, --> I think your hypothesis is that EDs and their related symptoms are more prevalent in the obese populations that are also exhibiting neurodevelopmental problems like ADHD and ASD. I think this should still be a clearer objective and the need to explain why previous work has not already shown this should be more explained.
line 108: Are these 58 declinations from the 76 that were recruited, or is this excess of 76 that participated?
line 114: This line sounds like results
line 118: The sentence "The definition of LOC eating..." being based on a question is incorrectly worded. A definition should not be based on the question. Perhaps you mean that LOC eating is measured by asking ____ with ___ number of items.
line 126: Was there a reason to administer the questionnaires to two different groups of kids? What was the reasoning behind this and if not, why did this occur?
line 141: You reported these numbers at the beginning of the other scale's explanation whereas this was reported at the end. Both sound like results and should be saved until the results section.
You may also consider adding why you used 2 scales instead of one.
line 146: Here you still have not defined the DSW-5 and you use roman numerals instead of numbers. Be consistent and be sure to define it.
line 147: You might consider a comma after "EDs"
line 153: I believe 'feeling of disgust or guilt' should be 'feelings of disgust or guilt'
line 151-155: This sentence is too long. It makes it difficult to understand and follow.
line 154: If you could not assign BED diagnoses, how do you defend the use of this method?
line 169: Better explain the use of the different chi-square test and Fisher's test. The "(one cell with less than five") is not enough explanation and makes it confusing.
line 174: You might consider an introductory sentence summarizing your results.
Table 1: This table is confusing for a couple of reasons, (1) consider superscripts to define the acronyms line by line, (2) Use less words in the column headings, (3) you have spelled out 'binge eating disorder' more than one time, crowding your footnotes. I am not sure that is necessary even once since it is defined in text prior to the results, (4) I think this table could be reorganized and perhaps be split into 2 or 3 different tables.
line 197: What is according to the DSM-5? If it is just the acronym, I think a small citation would be fine. It is confusing to be referring to the DAWBA and the DSM-5 that wasn't also used with the DAWBA.
line 201: Consider commas around "Shape concern and Weight concern."
line 206: one should be '1.' It is my understanding that if the numbers have units they can be written as numbers.
Table 2: Does ' mean (SD) ' suggest the data is written as mean +/- SD? SD to me means standard deviation, but it is written as if you are saying 'SD' stands for the mean of the data. Then you can take out the footnote.
Consider shortening the Table 2 title: "EDE-Q results expressed as mean +/- standard deviation (SD) in obese adolescents 12-17 years old by gender and age"
Consider centering and overall cleaning up table 2 data.
Consider reducing column title "individuals above cutoff"
After Table 2, before Table 3 (no line numbers here), "scored mostly in line with the norms" is very vague terminology
Table 3: It appears that you may have room to write the subscales out. I suggest doing this because it will make interpretting the table much simpler for the reader. I suggest the same for Table 4.
The line numbers restarted after Table 4.
line 11" 'those WHO did not"
line 30-32: This sentence compares the data of adolescents and children to adults. This does not seem appropriate.
line 54: What is the EDI-2?
line 52-62: I am unsure how relevant this is to your discussion.
line 74: This study should be cited.
line 101: Strengths and limitations - This paragraph should likely be split into two: Limitiations and a result oritened conclusion underscoring the strengths of the research, including why it's relevant and noteworthy.
line 113: I do not understand the sentence "it should thus be borne in mind that..."
line 117: The study population being treatment refractory is a large bias and may bias results.
Round 2
Reviewer 2 Report
Line 50 (Abstract): Why is there a new paragraph? I think the abstract should be one entity.
Line 59: per-cent is one word: percent
Line 62-63: Make the new phrase the beginning of the sentence to read, "According to the 5th edition of the American Psychiatric Association's Diagnostic and Statistical Manual of Mental Disorders (DSM-5), traditional eating disorders (EDs)...
Line 67-70: Sentence beginning with "BED is rare in children..." is confusing and too long. Break it up into more than one sentence.
Line 74-75: "can contribute to the development of overweight" does not make sense. The development of overweight ___ ? children?
Line 78-79: Cite this 'study.'
Line 88-92: Once sentence should be broken into 2 sentences.
Line 101-104: This sentence should be more concise and less wordy. It is unclear in it's current state.
Line 114-116: Much better
Line 133: Start the sentence with "The..."
Line 134: You should state what the subscales are like you do in the next paragraph. Stay consistent in explanations of questionnaires. The amount of questionnaires can be confusing and consistency is key to keeping the reader engaged and aware of the methods/instruments.
Line 141: Start a new sentence after the quotation mark.
Line 145: It is again confusing if this is a new study group because of the way you have written a different age range - make sure it's clear that of the 76 total study group members, only the 10-17 year olds were including in the adminisitering of this questionnaire.
Line 147: You state the amount of items in this questionnaire but not for the one above. Why is this? Stay consistent with questionnaire explanations.
Line 162-163: Why is there extra space here?
Line 164-166: This sounds like an introduction to the instruments. Pehaps move to the beginning.
Line 191-193: Random space
Line 198-204: For sample populations you shoudl be using the standard error instead of standard deviation. It would be better to report and analyze the statistics with SE. A SD is the standard deviation of a population. A standard error is the deviation of a sample population.
Line 216: Table 1 is better. However, the title is still too wordy. "if not otherwise stated" sounds like a conversation or footnote phrase. This should not be in a title of a Table.
Table 1 may be better if it was split into three tables on anthropometric data, eating disorders and neurodevelopment diagnoses. This table is confusing to me too especially with the footnotes.
Line 239-242: Break this sentence up.
Line 253: sevent out of twelve individuals should be 7 out of 12 individuals
Line 259-262: Is the "divided by gender and age" a necessary phrase. Perhaps renaming " EDE-Q results expressed as mean +/- standard deviation (SD) in male and female adolescents (12-15 and 16-17 years old) with obesity. Better table!
Line 268: Are the 57 out of 60 within the total group of 76? Are 60 of the 76 in the 10-17 years old group? The groups can be confusing.
Line 276-279: Perhaps rename the title like the previous table to might it tighter and more concise without losing details - Table 4's title is much better - stay consistent for each table naming style - I suggest using Table 4 as a template.
Table 3 = MUCH BETTER
Table 4 = MUCH BETTER
You lost your line numbers - so I'll go by page numbers.
pg. 15: paragraph after Table 4 - Shouldn't this be in the introduction of Table 1? This seems out of place.
pg. 16: second paragraph starting with: "According to the DAWBA..." - this is a marginal finding and should perhaps be stated as such.
pg. 16 First paragraph of discussion: "The prevalence of Eds was 10.5%...." and the sentence directly after it are redundant. One should be eliminated. They are stating the same thing.
For the sentence following, "We further wanted to... " should it be 'for' LOC instead of 'with' LOC? Start a new sentence at However, the study...
pg. 18, first paragraph: what is the 'subjective condition' you reference in the middle. That is a bit confusing for your point.
pg. 18 second paragraph: Towards the end start a new sentence with "However..."
pg. 19: top of page: cite Ro and colleagues [_], and perhaps re-write for conciseness: Ro and colleagues [_] found an association with higher scores and younger women which may explain why this study's scores were much higher.
pg. 19: Of these limitations you need to separate into some paragraphs. Additionaly you need to refute your limitations with why your paper is still strong. Many of them are heavily undermining your work. For ex. in your 3rd limitation, is this customary in your research type? If so, it should be stated why not having a control group is 'accetpable,' because otherwise it is not. Towards the end of the paragraph, "In respect of the EDI-C" is a phrase I do not understand.
pg. 20: In the middle of the first paragraph, the phrase "is probably the currently most frequently" is used. Please rephrase for clarity.
pg. 20, last line: "both assessed by experts'' instead of expertise, maybe?
pg. 21: Should this paragraph be set apart as the conclusion?
Author Response
Editors-in-Chief Prof. Dr. Lluis Serra-Majem and Prof. Dr. Maria Luz Fernandez
Special Issue Editor Prof. Dr. Anja Hilbert
Manuscript ID: nutrients-570519
Type of manuscript: Article
Title: Is there an overlap between eating disorders and neurodevelopmental disorders in children with obesity?
Dear Editors,
Regarding our manuscript titled “Is there an overlap between eating disorders and neurodevelopmental disorders in children with obesity?”, we have tried to make all the suggested changes based on reviewer #2’s comments after the first revision. All the amendments in the text have been highlighted in turquoise. Please find below the answers to the comments and questions.
Sincerely,
Elisabet Wentz Anna Björk Jovanna Dahlgren
MD, PhD, Professor MD, PhD Student MD, PhD, Professor
Corresponding author: Elisabet Wentz, Department of Psychiatry and Neurochemistry, Institute of Neuroscience and Physiology, University of Gothenburg, Högsbo Hospital, Tunnlandsgatan 12 A, SE-42138 Västra Frölunda, Sweden. email: elisabet.wentz@gu.se
|
Reviewer #2 |
Authors’ replies |
|
|
Thank you for reviewing our manuscript a second round and giving us constructive comments how to improve the manuscript.
|
|
Line 50 (Abstract): Why is there a new paragraph? I think the abstract should be one entity.
|
The abstract is now one entity. |
|
Line 59: per-cent is one word: percent |
Percent is now one word (the track changes in the pdf file unfortunately gave the impression that we hadn’t amended this comment in the previous review) (page 4)
|
|
Line 62-63: Make the new phrase the beginning of the sentence to read, "According to the 5th edition of the American Psychiatric Association's Diagnostic and Statistical Manual of Mental Disorders (DSM-5), traditional eating disorders (EDs)... |
We have rephrased the sentence according to reviewer #2’s suggestion (page 4, second paragraph) |
|
Line 67-70: Sentence beginning with "BED is rare in children..." is confusing and too long. Break it up into more than one sentence. |
We have broken up the sentence into three sentences (page 4, second paragraph). |
|
Line 74-75: "can contribute to the development of overweight" does not make sense. The development of overweight ___ ? children? |
We have now adjusted the sentence. |
|
Line 78-79: Cite this 'study.' |
The study has now been cited (page 5). |
|
Line 88-92: Once sentence should be broken into 2 sentences.
|
We have broken the sentence into two sentences (page 5, second paragraph). |
|
Line 101-104: This sentence should be more concise and less wordy. It is unclear in it's current state.
|
On page 5, last paragraph, the sentence has been adjusted in order to make it clearer. |
|
Line 114-116: Much better
|
Thanks. |
|
Line 133: Start the sentence with "The..." |
Now adjusted. |
|
Line 134: You should state what the subscales are like you do in the next paragraph. Stay consistent in explanations of questionnaires. The amount of questionnaires can be confusing and consistency is key to keeping the reader engaged and aware of the methods/instruments. |
On page 7, first paragraph, we have added the names of the four subscales. |
|
Line 141: Start a new sentence after the quotation mark. |
Now adjusted. |
|
Line 145: It is again confusing if this is a new study group because of the way you have written a different age range - make sure it's clear that of the 76 total study group members, only the 10-17 year olds were including in the adminisitering of this questionnaire.
|
On page 7, last paragraph, this has been clarified.
|
|
Line 147: You state the amount of items in this questionnaire but not for the one above. Why is this? Stay consistent with questionnaire explanations.
|
See page 7, first paragraph, where we have made the amendments. |
|
Line 162-163: Why is there extra space here?
|
The extra space is now removed. |
|
Line 164-166: This sounds like an introduction to the instruments. Pehaps move to the beginning.
|
We welcome this suggestion and have moved the paragraph to the top of the Instrument section (page 6, last paragraph). |
|
Line 191-193: Random space
|
Now removed. |
|
Line 198-204: For sample populations you shoudl be using the standard error instead of standard deviation. It would be better to report and analyze the statistics with SE. A SD is the standard deviation of a population. A standard error is the deviation of a sample population.
|
We are grateful for this important comment. The literature pertaining to the EDE-Q and the EDI-C that we are referring to has reported their findings using standard deviation (SD) (based on populations as well as sample populations) and therefore we consider it more suitable to stick with SD in this manuscript. |
|
Line 216: Table 1 is better. However, the title is still too wordy. "if not otherwise stated" sounds like a conversation or footnote phrase. This should not be in a title of a Table.
|
Regarding Table 1, the title has been shortened. |
|
Table 1 may be better if it was split into three tables on anthropometric data, eating disorders and neurodevelopment diagnoses. This table is confusing to me too especially with the footnotes.
|
The table has now been split into two tables; Table 1 and Table 2. There were unfortunately not enough data to split into three tables. |
|
Line 239-242: Break this sentence up.
|
We have split the sentence into two. |
|
Line 253: sevent out of twelve individuals should be 7 out of 12 individuals
|
We are now using numbers instead of letters (page 12). |
|
Line 259-262: Is the "divided by gender and age" a necessary phrase. Perhaps renaming " EDE-Q results expressed as mean +/- standard deviation (SD) in male and female adolescents (12-15 and 16-17 years old) with obesity. Better table!
|
We are happy to see that reviewer #2 finds Table 3 (former Table 2) improved. The title of Table 3 (former Table 2) has been changed according to the recommendations. |
|
Line 268: Are the 57 out of 60 within the total group of 76? Are 60 of the 76 in the 10-17 years old group? The groups can be confusing.
|
We have now clarified that they belong to the study group (n=76) (page 13). |
|
Line 276-279: Perhaps rename the title like the previous table to might it tighter and more concise without losing details – Table 4’s title is much better as a template. Table 3 = MUCH BETTER Table 4 = MUCH BETTER
|
The table title is now renamed.
We are glad that reviewer #2 is content with Table 3 and 4 (now renamed to Table 4 and Table 5). |
|
You lost your line numbers - so I'll go by page numbers. pg. 15: paragraph after Table 4 - Shouldn't this be in the introduction of Table 1? This seems out of place.
|
We agree with reviewer #2 and therefore we have moved the section with the title “ED symptoms and correlations with other parameters including ADHD and ASD” to page 11 |
|
pg. 16: second paragraph starting with: "According to the DAWBA..." - this is a marginal finding and should perhaps be stated as such.
|
We have minimized the text regarding “major depression” (now on page 11).
|
|
pg. 16 First paragraph of discussion: "The prevalence of Eds was 10.5%...." and the sentence directly after it are redundant. One should be eliminated. They are stating the same thing.
|
In the first sentence we have added that we mean BED and OSFED-BED when using the term “EDs”, In the sentence directly after we are referring to LOC eating, exclusively (page 15). |
|
For the sentence following, "We further wanted to... " should it be 'for' LOC instead of 'with' LOC? Start a new sentence at However, the study...
|
In the first paragraph of the Discussion section we have changed/clarified this sentence. A new sentence has been created starting with “However,….” |
|
pg. 18, first paragraph: what is the 'subjective condition' you reference in the middle. That is a bit confusing for your point.
|
This is now explained on page 16, last paragraph. |
|
pg. 18 second paragraph: Towards the end start a new sentence with "However..."
|
We have started a new sentence with “However,…..” (now on page 17) |
|
pg. 19: top of page: cite Ro and colleagues [_], and perhaps re-write for conciseness: Ro and colleagues [_] found an association with higher scores and younger women which may explain why this study's scores were much higher.
|
We have changed the sentence in accordance with reviewer #2’s recommendations (page 17). |
|
pg. 19: Of these limitations you need to separate into some paragraphs. Additionaly you need to refute your limitations with why your paper is still strong. Many of them are heavily undermining your work. For ex. in your 3rd limitation, is this customary in your research type? If so, it should be stated why not having a control group is 'accetpable,' because otherwise it is not. Towards the end of the paragraph, "In respect of the EDI-C" is a phrase I do not understand.
|
We are now stating why not having a control group is acceptable. The third and fourth limitation has now been put together. We have refuted the third and the fourth limitation (page 18) and the final limitation (page 19).
The sentence pertaining to the EDI-C has now been amended (page 18). |
|
pg. 20: In the middle of the first paragraph, the phrase "is probably the currently most frequently" is used. Please rephrase for clarity.
|
We have clarified/rephrased the sentence (page 19, first paragraph) |
|
pg. 20, last line: "both assessed by experts'' instead of expertise, maybe?
|
On page 20 we have changed “expertise” to “experts”. |
|
pg. 21: Should this paragraph be set apart as the conclusion?
|
A paragraph with the heading “Conclusions” has now been added. |
